# A Narrative Review of Surgery for Prolactinomas: Considerations and Controversies

**DOI:** 10.3390/jcm14041089

**Published:** 2025-02-08

**Authors:** Jennifer A. Mann, Yves Starreveld, Jay Riva-Cambrin, Kirstie Lithgow

**Affiliations:** 1Department of Clinical Neurosciences, Section of Neurosurgery, University of Calgary, Calgary, AB T2N 1N4, Canada; jennifer.mann@ucalgary.ca (J.A.M.); ystarrev@ucalgary.ca (Y.S.);; 2Department of Community Health Sciences, University of Calgary, Calgary, AB T2N 1N4, Canada; 3Division of Endocrinology and Metabolism, Department of Medicine, University of Calgary, Calgary, AB T2N 1N4, Canada; 4Hotchkiss Brain Institute, University of Calgary, Calgary, AB T2N 1N4, Canada

**Keywords:** dopamine agonist, pituitary adenoma, prolactinoma, transsphenoidal surgery

## Abstract

For several decades, dopamine agonist therapy has been the mainstay of treatment for prolactinomas, with surgery generally considered a second line for cases failing medical therapy due to intolerance or resistance. There is increasing recognition of the burden of long-term DA therapy; many patients experience debilitating side effects, and emerging evidence demonstrates that the prevalence of impulse control disorders has been vastly underreported. Long-term DA therapy is associated with significant costs to patients and healthcare systems, which is projected to exceed that of surgery in many circumstances. Recent advancements in surgical approaches, including endoscopic transsphenoidal surgery, have led to improved surgical outcomes (82–100% remission rates; serious complication rates < 2%), prompting a reappraisal of the role of surgery for prolactinoma. Favourable surgical outcomes have been observed in both remission and complication rates for microprolactinomas and well-circumscribed macroprolactinomas, leading to consideration of surgery as an earlier, or first-line, option in the treatment paradigm. Potential advantages of surgical management should be weighed against institutional case volume and expertise, the risk of perioperative complications, and the need for adjuvant medical therapy post-operatively. Ultimately, patients and care-providers should engage in shared decision-making following informed discussion about the risks and benefits of both medical and surgical approaches.

## 1. Introduction

Prolactinomas are the most common subtype of pituitary adenoma, comprising over half of all pituitary adenomas and presenting most commonly in women of reproductive age [1]. At presentation, they are classified based on maximum dimensions: macroprolactinomas are >1 cm and microprolactinomas are <1 cm [2]. Patients can present with signs and symptoms of mass effect (most commonly visual field deficits), hypopituitarism (hyposecretion of one or more pituitary hormones), and hyperprolactinemia [3,4]. The features of hyperprolactinemia can occur either due to the impact of hypogonadism (due to inhibition of pulsatile gonadotropin-releasing hormone (GnRH) secondary to hyperprolactinemia) or from the primary effects of elevated prolactin [3]. Manifestations of hypogonadism can include decreased libido, amenorrhea/oligomenorrhea, impaired erectile or ejaculatory function, infertility, and decreased bone mineral density [3]. Manifestations specific to hyperprolactinemia include galactorrhea (most common in reproductive-age women) and headache. Diagnosis of prolactinoma requires the presence of a sellar lesion consistent with a pituitary adenoma in conjunction with elevated serum prolactin (typically >250 μg/L with elevation in prolactin in proportion to adenoma size) [2]. The impact of medications known to cause elevations in prolactin should be considered when interpreting prolactin results [3]. 

Dopamine agonists (DAs) are derived from ergot, a group of fungi that grow on rye and related plants [5]. In 1676, it was observed that ergot inhibited lactation [6]. Three hundred years later, the first report of a reduction in the size of two prolactinomas following DA therapy with a semisynthetic ergot alkaloid (bromocriptine) was published [7]. In 1981, a prospective study of eight patients with macroprolactinomas demonstrated reduction or normalization of serum prolactin as well as regression in tumour size following three months of bromocriptine treatment [8]. The proven anti-secretory and antiproliferative potential of bromocriptine led to it replacing surgery for the first-line management of prolactinomas in the years that followed [9]. Medical management of prolactinomas was further improved following the approval of the DA cabergoline in the 1990s, which demonstrated superior efficacy and tolerability over bromocriptine [10] due to selectivity for D2R receptors [3]. Currently commercially available DAs include bromocriptine, cabergoline, and quinagolide [3]. Synthesis of results from previous studies have demonstrated that DA therapy leads to tumour shrinkage in 20 to 100% (median 62%) and normalization of serum prolactin in 40 to 100% (median 68%) [2]. The greatest efficacy is reported to occur within the first six months of treatment initiation [11], and normalization of prolactin and tumour volume reduction exceeding 25% within the first 3 months of treatment is predictive of long-term response [12,13]. In select cases (i.e. normal prolactin and no visible tumour on MRI), withdrawal of DA after two or more years of therapy can be attempted with observed remission rates of 26 to 69% [2,3,14]. 

While DA therapy has been recommended for the first-line management of prolactinomas by international society guidelines for many years [2,14,15], advances and longitudinal experience with endoscopic endonasal surgery have yielded improved surgical outcomes over the last two decades [16], prompting reappraisal of the role of surgical management for prolactinoma [13,17]. Surgery on the pituitary gland in humans began in the late 19th century, led by Harvey Cushing, with the primary indications being vision loss and headache [18]. Transcranial approaches (requiring craniotomy with an incision through the overlying tissues) were utilized by Cushing and subsequent surgeons throughout the 1990s [18]. In the early 20th century, as otolaryngologists began to use the endoscope in the nasal cavity, neurosurgeons began to trial transsphenoidal approaches to the pituitary [18]. The first transsphenoidal approach is believed to have been performed by Austrian surgeon Herman Schloffer in 1907 and was performed via a lateral rhinotomy [19]. The transsphenoidal approach, in which surgeons access the sphenoid sinus to approach the sella, was largely abandoned and transcranial approaches were preferred until it was popularized by Jules Hardy in the 1960s who pioneered the use of the operating microscope and fluoroscopy [19,20]. Hardy had spent a year training with French surgeon Gerard Guiot, who was the first to use the endoscope in transsphenoidal surgery, although the technique did not gain traction until the 1990s when the endoscope was optimized by Karl Storz for visualization of anatomic structures [18,19,21,22,23]. Following the advent of the endoscopic transsphenoidal approach in the early 1990s [19,22,23], modern neurosurgeons have refined both microscopic and endoscopic endonasal transsphenoidal approaches to pituitary tumours, aided significantly by increased collaboration with otolaryngologists and the use of image guidance [18,24]. The contemporary microsurgical transsphenoidal approach utilizes instruments passed through the nostrils with a Hardy speculum facilitating visualization through the operative microscope [20]. Endoscopic transsphenoidal surgery (TSS) is a minimally invasive technique by which surgeons access the sella through the sphenoid sinus with instruments passed through the nostrils and anatomic visualization through the endoscope, allowing for increased anatomic visualization as compared to the microscopic approach, often performed by a neurosurgeon and otolaryngologist team [20]. 

Potential advantages conferred by endoscopic TSS include decreased complication rates (due to improved anatomic visualization) and increased endocrinological remission rates [25,26]. Surgical remission rates are pooled at 82–100% in endoscopic series versus 71–93% in microscopic surgery [27]. Furthermore, operative time, length of hospital stays, and blood loss are reduced with endoscopic TSS, making this approach highly safe and effective [28,29]. A systematic review and meta-analysis reported long-term postoperative remission rates of 83% for microprolactinomas (19 studies, n = 354) and 60% for macroprolactinomas (17 studies, n = 639) [17]. Reported severe post-operative complication rates are <2%, making surgical resection appealing for amenable lesions [27]. Accordingly, the most recent Pituitary Society International Consensus Statement recommends surgery as a first-line alternative to DA therapy in select cases [13] (Figure 1). 

To identify relevant literature concerning the surgical management of prolactinoma, we performed a search of PubMed including the following MeSH terms: prolactinoma, surgery, neurosurgery, and neurosurgical procedure. We additionally utilized the reference list review of the included manuscripts and known literature in the authors’ respective fields based on expertise. In this review, we will summarize the traditional indications for surgical management of prolactinoma before reappraising the contemporary role of surgery with emphasis on the expanding role of surgery as a first-line intervention. We have highlighted changes from previous 2011 guidelines [2] to the recent 2023 guidelines from the Pituitary Society [13] pertaining to the surgical management of prolactinoma. Finally, we will review perioperative considerations and emphasize the necessity of shared decision-making between healthcare providers and patients with prolactinoma. 

## 2. Indications for Surgery

### 2.1. Traditional Indications for Surgery 

Traditional indications for the surgical management of prolactinoma include resistance or intolerance to DA therapy, neurological deficit, pituitary apoplexy, cerebrospinal fluid (CSF) leak, and for volume reduction prior to pregnancy [2,30]. Within each of these indications, there are important nuances that warrant multidisciplinary discussion and shared decision-making with patients.

### 2.2. Resistance to Dopamine Agonist Therapy

The accepted definition of DA resistance is failure to achieve normoprolactinemia with maximally tolerated doses of DA and failure to achieve a 50% reduction in tumour size [2,31]. Resistance to cabergoline has been reported at 10% for micro- and 18% for macroprolactinomas [32]. However, it should be emphasized that not all patients who meet the definition of DA resistance will require surgery. In the real-world treatment of macroprolactinomas, the most important endpoint is resolution of mass effect (i.e. decompression of optic chiasm), and once this is achieved, many patients can be stabilized on long-term DA despite not achieving normoprolactinemia or 50% size reduction [33]. For microprolactinomas, size reduction is not a clinically important outcome, and therefore, decisions around surgical management for resistant cases will consider factors such as long-term treatment burden, side effects from escalating doses of DA, and symptoms of hyperprolactinemia. The presence of DA resistance alone is therefore not an absolute indication for surgery unless there are mass effect concerns such as chiasmal compression. 

Observational studies examining post-operative remission rates in patients previously treated with DA demonstrate lower remission rates than primary surgery [34,35]; however, caution is warranted in interpreting these findings given differences in the baseline characteristics of cases that undergo primary vs. second-line surgery. A systematic review and meta-analysis (10 studies; n = 657) reported that 38% of patients who had surgery after failing DA therapy achieved long-term remission, and 62% achieved remission with multimodal therapy (additional surgery, radiosurgery, or DA reinitiation) [34]. Subgroup analysis suggested that surgery alone had no effect on the long-term remission of macroprolactinomas and led to remission in 66% of microprolactinomas; however, 43% of macroprolactinomas achieved remission on multimodal therapy [34]. It is important to consider that even if post-operative remission is not achieved, operative intervention may facilitate improved biochemical control with lower DA doses [34,35].

### 2.3. Dopamine Agonist Intolerance 

DA intolerance is roughly defined as “side effects of the medication precluding its use [2]”. Common side effects of DA are dose-related and include gastrointestinal upset, dizziness, headache, and fatigue, which can be persistent and detrimental to quality of life [13,36,37]. Switching to cabergoline can be trialed in patients who do not tolerate bromocriptine due to its more favourable side effect profile [2]. The side effects of DA are dose-related, with more frequent adverse effects occurring with escalating doses [2]. Per a large retrospective cohort study of 455 patients taking cabergoline (median maintenance dose 0.5 mg weekly), 8.5% of patients experienced minor adverse effects and 4% had major or persistent side effects, with headache, postural hypotension, nausea, and sleepiness being the most commonly reported [38]. A systematic review and meta-analysis including both surgically and medically managed prolactinomas reported higher pooled rates of side effects including fatigue (30%), libido changes resulting from side effects (28%), sleep disorders (25%), and nausea (17%), presumably related to higher doses of DA in this cohort [17].

Rare but serious potential side effects of DA include neuropsychiatric issues such as mood changes or impulse control disorders (ICDs), including hypersexuality, pathological gambling, compulsive shopping, and compulsive eating [13,39,40]. A previous systematic review reported pooled percentages of 3% for ICDs in prolactinoma patients treated with DA [17]. However, a cross-sectional multicentre study that assessed 308 patients with DA-treated prolactinoma using validated questionnaires reported ICD prevalence of 17% [41]. These clinically important adverse effects have been previously underreported due to a lack of clinical recognition and/or formal assessment [42]. It is crucial that care providers disclose and monitor for these effects which may have devastating consequences for patients. 

### 2.4. Neurological Deficit, Pituitary Apoplexy, and CSF Leak 

Macroprolactinomas may cause visual dysfunction at presentation due to suprasellar extension and compression of the optic apparatus [4]. However, even large and invasive (e.g., tumour extension into the cavernous sinus) macroadenomas may respond rapidly to DA treatment with resolution of mass effect and improvement in vision [4]. If a trial of DA therapy is undertaken in this setting, this should involve a multidisciplinary approach with endocrinology, ophthalmology, and neurosurgery. Short-term reassessment of visual function following DA initiation is critical and surgical management should be offered if it does not improve [4].

Pituitary apoplexy is an emergent complication that presents with symptoms including headache, visual dysfunction, altered level of consciousness, and pituitary hormone deficiency due to infarction or hemorrhage of a pituitary adenoma [43]. Management of pituitary apoplexy is individualized, and cases with severe visual and/or neurological dysfunction are offered surgical management while milder cases can be managed conservatively [4,43].

In large invasive macroprolactinomas, DA therapy can rarely cause rapid tumour shrinkage, leading to spontaneous CSF leak [13,44]. Patients at risk should be counselled about signs and symptoms (e.g., clear nasal discharge) and surgical treatment is usually required if this complication occurs [13,45].

### 2.5. Volume Reduction Prior to Pregnancy

Previous guidelines recommend patients with macroprolactinomas who do not achieve tumour shrinkage with DA be counselled about the potential benefit of surgery before attempting pregnancy [2], as the risk of symptomatic mass enlargement is estimated at 21% [46]. If surgery is offered for this indication, it is crucial that patients are informed about the risk of acquired hypopituitarism following surgery, which would necessitate use of assisted reproductive technologies for conception as well as lifelong hormone replacement therapy [2].

### 2.6. Reappraisal of Surgery for Prolactinomas 

The 2023 Pituitary Society International Consensus Guidelines highlight an emerging role for first-line surgery in primary management for select prolactinomas due to favourable surgical outcomes, the treatment burden of long-term DA therapy, and patient preference. The select cases for consideration of first-line surgical intervention as emphasized in the consensus statement include the following: microprolactinomas, well-circumscribed (Knosp grade 0 and 1) macroprolactinomas, young women, and patients desiring pregnancy [13]. 

Initial operative approaches to the pituitary were via transcranial routes; experimentation with subfrontal, transfrontal, subtemporal, and transtemporal routes eventually gave way to the pterional approach which is favoured in modern neurosurgery, whereby surgeons create a frontotemporal boneflap centred over the depression of the sphenoid ridge to access the basal arachnoid cisterns through the Sylvian fissure [18]. Pterional craniotomy retains utility as a surgical approach to prolactinoma in instances such as failed transsphenoidal surgery, extrasellar tumour extension, or patient-specific anatomic limitations [20]. The early transsphenoidal approach was initially performed transnasally, requiring incisions on or around the nose, reflection of the nose, and various degrees of resection involving the maxillary, frontal, or ethmoid sinuses [19,47]. Sublabial approaches were also popular for transsphenoidal surgery throughout the 1900s, involving resection of the bony nasal septum and nasal turbinate(s) [19]. A myriad of additional approaches have been trialed to access the sella throughout the 19th, 20th, and 21st centuries, which are outside of the scope of this manuscript [19,47]. The modern transsphenoidal approach is performed endonasally with the aid of the operating microscope or endoscope. Given the trajectory of the endoscopic endonasal approach, there are inherent limitations, including suprasellar or lateral tumour extension, brain invasion, and tumour consistency, in which instances an open craniotomy may be more favourable [48]. In addition to standard transsphenoidal approaches, surgeons have developed extended transsphenoidal approaches, maximizing access to the skull base to aid in the resection of invasive tumours and providing a corridor to the cavernous sinus [47,49]. Combined approaches are another viable option for more invasive tumours, such as the endoscopic transorbital approach (which provides further reach into the cavernous sinus in a minimally invasive manner) in combination with endoscopic endonasal surgery [50].

Important considerations include careful case selection and the availability of surgical expertise. With regard to case selection, tumour size and invasiveness are of paramount importance. The Knosp grade is a magnetic resonance imaging (MRI)-based classification scheme for tumoural invasion of the cavernous sinus, ranging from 0 (normal) to 4 (complete encasement of the internal carotid artery), with non-invasive adenomas generally considered < grade 2 and invasive adenomas considered ≥ grade 2 (Table 1) [51]. Microprolactinomas and well-circumscribed macroprolactinomas (Knosp grades 0 and 1) have favourable remission rates in the hands of experienced neurosurgeons, making surgery attractive [13]. For more invasive prolactinomas, surgical remission is often not feasible, and additional treatment post-operatively (adjuvant therapy) with either ongoing DA or radiotherapy is required. However, even if complete remission is not achieved, surgery can lead to a clinically significant reduction in prolactin levels, such that lower doses of DA can be used to control hyperprolactinemia [34,35].

A retrospective study of 86 Knosp ≤ 1 prolactinomas (41 macro- and 45 microprolactinomas) treated with first-line surgery demonstrated long-term remission (median follow-up: 80 months) of 72% for micro- and 45% for macroprolactinomas [52]. However, 24% of the micro- and 49% of the macroprolactinomas (notably, 76% of Knosp 1 vs. 29% of Knosp 0) required adjuvant DA to achieve remission [52]. Another series of 78 prolactinomas (mostly pre-treated with DA) reported long-term remission rates (mean follow-up: 66 months) of 65% for Knosp 0–2 tumours, which increased to 81% when including remission with adjuvant DA [53]. A previous systematic review and meta-analysis (13 studies and 809 patients) demonstrated significantly higher remission rates for patients treated with surgery than with DA (88% vs. 52%; *p* = 0.001) [54]. Additionally, remission rates of 91 vs. 60% (*p* = 0.002) for micro- and 77% vs. 43% (*p* = 0.003) for macroprolactinomas were reported for first-line surgical vs. medical management, respectively (although this study was unable to assess the use of post-operative DA) [54]. 

Reported complication rates following prolactinoma surgery are favourable. A systematic review and meta-analysis including surgically managed micro- and macroprolactinomas (25 studies; n = 1836) reported 0% mortality, severe complications (permanent diabetes insipidus, meningitis, and CSF leak) ≤ 3%, and anterior hypopituitarism in 2% [17]. A retrospective cohort study of 114 microprolactinomas reported an overall complication rate of 4% (of which 3% were epistaxis, rhinitis, and sinusitis), with 0% hypopituitarism and only a single case of permanent diabetes insipidus [55]. Despite these promising data, post-operative remission and complication rates reported in the literature are primarily from tertiary referral centres with expert skull base surgeons; therefore, these results are only generalizable to similar care settings. Furthermore, such centres may still lack recent experience with operative management of prolactinomas and therefore require an additional case volume to optimize outcomes. Abou-Al-Shaar et al. [53] reported significantly improved long-term remission rates for Knosp 0–2 prolactinomas in the second decade of experience (77% in 2011–2019 vs. 47% in 2002–2010), suggesting a surgical learning curve. The role of surgery in the treatment paradigm of prolactinomas should therefore be tailored to institutional case volume and surgical expertise. 

Patients with prolactinoma experience worse quality of life, particularly mental health, than healthy controls, and mental and physical quality of life scores vary inversely with prolactin levels, emphasizing the importance of assessing patient-reported outcome measures in the surgical decision-making process [56]. In a recent prospective cohort study, Van Trigt et al. [57] assessed health-related quality of life with the Leiden Bother and Needs Pituitary questionnaire in 100 prolactinoma patients treated via endoscopic TSS and found that Bother and Needs scores both decreased significantly post-operatively. In a recent systematic review of 18 articles, a clear difference in quality of life was not found between medical or surgical treatment modality, but surgery was associated with improved quality of life as soon as 5 days postoperatively, with continuous improvement throughout the first year, with prolactinoma showing the greatest quality of life improvement of all adenoma types [56]. Although further study is needed to better understand the quality of life outcomes in medically versus surgically treated prolactinoma patients, these results support the immediate positive effects of surgery in this patient population. 

The cost-effectiveness of surgery vs. long-term DA therapy warrants consideration [13], given that the majority of patients will require life-long medical therapy [17]. Two previous analyses have demonstrated surgery is more cost-effective than medical therapy for microprolactinomas [58,59], which is amplified with cumulative years of medication use [58]. However, the cost–advantage switched to favour cabergoline over surgical intervention only when the theoretical surgical cure rate fell below 30% [59].

Patient preference is becoming an important indication for surgical intervention, reflecting the emphasis on patient-centric care in modern neurosurgery [60]. In particular, surgery may be a favourable option for young women with micro- or Knosp 0-1 prolactinomas to restore fertility and avoid long-term DA use [13]. A meta-analysis of 14 studies (n = 603) including women undergoing TSS for prolactinoma reported a significant reduction in rates of amenorrhea (96% vs. 40%; *p* < 0.01) and galactorrhea (84% vs. 29%; *p* < 0.01) post-operatively [61]. Rates of hypopituitarism could not be analyzed due to heterogeneity but ranged from 0–23%, with rates >10% reported in only two studies [61], which further highlights the importance of institutional expertise. We suggest that care providers engage in shared decision-making (SDM) with patients to emphasize patient preference. SDM should encourage patient collaboration by presenting clearly the prolactinoma diagnosis and the risks and benefits of both treatment options (e.g., DA therapy versus surgery) [62]. Based on the authors’ experience, this is best accomplished in a multi-disciplinary setting, whereby a neurosurgeon and endocrinologist present the patient with respective treatment options, patient and support persons’ questions are addressed immediately, the patient’s opinion is explicitly elicited, and they are empowered to make the best treatment decision for their circumstances. SDM requires that physicians are attentive to the unique preferences and goals of the patient (e.g., do they desire fertility? If not now, will they in the future?), tailor the risk/benefit discussion accordingly, and provide time and space for the patient’s voice in the discussion [62]. 

## 3. Perioperative Considerations

A collaborative approach between endocrinology and neurosurgery is imperative in managing patients who undergo surgical management of prolactinoma. General considerations for the perioperative management of pituitary adenoma are beyond the scope of this review and are discussed in detail elsewhere [63]. Patients who opt to undergo primary surgical treatment of prolactinoma should be counselled regarding the risks and benefits of both surgical and medical approaches so that they can make an informed decision [13]. Surgery for prolactinoma should be performed at a pituitary centre of excellence [64] by an experienced pituitary surgeon [13]. 

As mentioned, surgical intervention for pituitary adenoma is not without risk of complications. The most serious potential complication of transsphenoidal pituitary surgery is carotid artery injury, which is exceedingly rare (<0.2% in large endoscopic transsphenoidal series [65]). Additional serious complications include CSF leak, hypopituitarism, permanent diabetes insipidus, and visual deterioration, which all remain rare. More common complications include transient diabetes insipidus and syndrome of inappropriate antidiuretic hormone (SIADH) secretion; these are readily treatable, short-term consequences (with rates ranging from 10 to 20% [66]). Intraoperatively, adenoma resection may require or cause surgical manipulation of the posterior pituitary lobe, stalk, or infundibulum, which can cause antidiuretic hormone (ADH) secretion disturbances [67]. ADH deficiency causes renal free water losses and subsequent hypernatremia, whereas ADH hypersecretion (SIADH) causes water retention and hyponatremia [67]. For both conditions, careful management of post-operative fluid and electrolyte balance is warranted, and management may be performed by neurosurgeons, although we suggest that endocrinologist involvement where possible is best practice. Diabetes insipidus most commonly occurs in the first two post-operative days and is often treated with the vasopressin analogue desmopressin (DDAVP), and SIADH typically occurs 4–10 days post-operatively and is commonly treated with fluid restriction [66,67]. Post-operative CSF leak (incidence <5% in experienced centres [68]) is managed by the neurosurgical team and treated via conservative management (e.g., bedrest), lumbar drain insertion, or revision surgery [67]. Post-operative adrenal insufficiency must be excluded with serum cortisol assessment and glucocorticoid therapy initiated in patients when appropriate [66].

Controversy exists about technical surgical challenges arising from tumour fibrosis secondary to pre-surgery treatment with DA [69,70]. Tumour texture limits the surgeon’s ability to utilize gentle suction and ring curettes for tumour resection and often adheres the tumour tissue to surrounding structures, which may increase the likelihood of incomplete resection and bleeding. A recent retrospective study including 290 prolactinomas (199 macroprolactinomas and 91 microprolactinomas) compared perioperative outcomes and pathological findings for prolactinomas that were pre-treated with DA vs. those that underwent primary surgery [70]. DA-pre-treated macroprolactinomas were found to have significantly higher intraoperative blood loss, longer surgical duration, and more overall surgical morbidity [70]. The authors demonstrated that DA pre-treatment was an independent risk factor for tenacious tumour consistency after adjustment for age, sex, tumour volume, and disease duration [70]. Pathological analysis illustrated increased intratumoural collagen content in the DA pre-treated cases, supporting tumour fibrosis as the putative mechanism [70]. Furthermore, a significant positive correlation between cumulative DA dose and intratumoural collagen content was observed for macroprolactinomas [70]. Taken together, these findings support that DA pre-treatment for macroprolactinomas may impact perioperative outcomes, and therefore, there is rationale to minimize or avoid DA exposure in cases when primary surgical management is preferred. However, it should be noted that differences in perioperative outcomes and intratumoural collagen content were not observed for microprolactinomas, though lower post-operative remission rates in the DA-pretreated group (87% vs. 100% in the initial surgery group) were reported [70]. Importantly, the DA exposure in this cohort was exclusively bromocriptine [70]; therefore, it remains unclear whether these findings are generalizable to the use of cabergoline, especially given that a previous study suggested that prolactinoma fibrosis is less pronounced in cabergoline than in bromocriptine [71].

Following surgical management, ongoing follow-up with endocrinology is required to determine if remission has been achieved. In cases where DA resistance is the surgical indication, a reduced dose may be efficacious following surgical debulking [72]. Ongoing follow-up of prolactin levels and titration of DA post-operatively should be performed by the endocrinologist [13], in addition to the usual post-operative biochemical monitoring that is standard of care for all pituitary adenomas [63].

## 4. Conclusions

Both traditional and contemporary indications for the surgical management of prolactinoma should consider patient, tumour, surgical, and system factors (Figure 2). Goals of surgery (remission for micro- and select macroprolactinomas vs. debulking for more invasive tumours) and potential risks and limitations should be clearly communicated to patients. Additional studies are required to further elucidate the impact of DA (cabergoline in particular) pre-treatment on tumour fibrosis and surgical remission rates to determine whether up-front DA should be routinely avoided for select cases until surgical options are discussed. Unique cases such as young females with microprolactinoma require particular emphasis on surgical options due to the potential for fertility restoration and sparing of DA therapy, which in turn could facilitate thousands of dollars in cost savings for the patient and healthcare system. Management decisions should ideally be made in a multidisciplinary care setting (with availability of both neurosurgical and endocrinological expertise) utilizing a patient-centred approach.

## Figures and Tables

**Figure 1 jcm-14-01089-f001:**
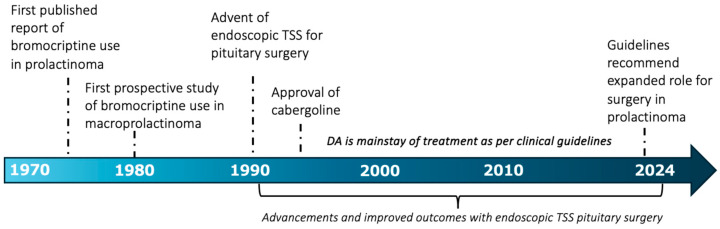
History and evolution of prolactinoma management. This figure is original to this submission so no credit or license is needed.

**Figure 2 jcm-14-01089-f002:**
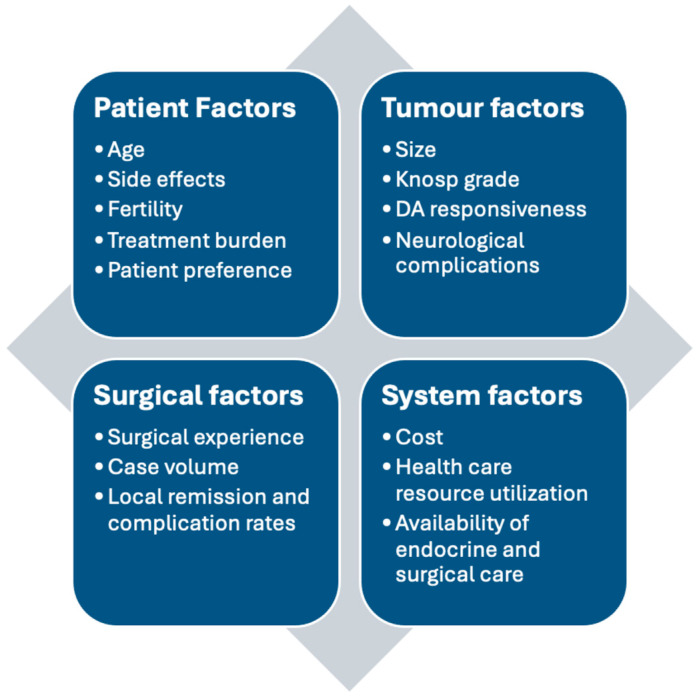
Factors impacting shared decision-making for the management of prolactinoma. This figure is original to this submission so no credit or license is needed.

**Table 1 jcm-14-01089-t001:** Knosp grade, adapted from Knosp et al. Classification of pituitary adenoma based on midsella magnetic resonance imaging. ICA: internal carotid artery.

Knosp Grade	Definition
0	No cavernous sinus invasion, the lesion does not extend past medial aspects of the intra- and supracavernous ICA
1	The lesion extends to, but not past, the tangent line through cross-sectional centres of the intra- and supracavernous ICA
2	The lesion extends to, but not past, the tangent line through lateral aspects of intra- and supracavernous ICA
3	The lesion invades past the tangent line through the lateral intra- and supracavernous ICA into the cavernous sinus
4	The lesion completely encases the intracavernous carotid artery and the cavernous sinus is invaded

## Data Availability

Not applicable.

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
