# Peer review of "A Narrative Review of Surgery for Prolactinomas: Considerations and Controversies"

_jcm, 2025, doi:10.3390/jcm14041089_

Round 1
Reviewer 1 Report
Comments and Suggestions for Authors
Dear authors,
Compliments for this interesting article; the surgical indications for prolactin-secreting adenomas have undergone many changes over time, and this chronology is well illustrated in the paper. There are some limitations that must necessarily be managed first
1) It is true that this is a comprehensive literature review, however, the literature search criteria should be stated
2) several times in the text the update made by the current guidelines is indicated, the latter should be better highlighted in a selected paragraph.
3) in the paragraph on surgery as the first line of treatment, the categories of patients who would benefit from it according to the current guidelines (in addition to women of childbearing age etc) should be better specified
4) this article should give a little more space to role of endoscopic endonasal EEA approach
I also recommend that you also refer to the fact that even more invasive tumors that move from the midline and go lateral can be attacked through combined endonasal and transorbital ETOA approaches, in this regard read and cite: https://doi.org/10.3390/jcm13092712
Author Response
Reviewer 1
Dear authors,
Compliments for this interesting article; the surgical indications for prolactin-secreting adenomas have undergone many changes over time, and this chronology is well illustrated in the paper. There are some limitations that must necessarily be managed first
- It is true that this is a comprehensive literature review, however, the literature search criteria should be stated
Thank you, we have added additional information about our search strategy in the introduction (page 4).
- several times in the text the update made by the current guidelines is indicated, the latter should be better highlighted in a selected paragraph.
We have added an additional statement (page 4, lines 111 to 113) clarifying that we are referring to the Pituitary Society 2023 guidelines vs. previous guidelines which were published in 2011.
- in the paragraph on surgery as the first line of treatment, the categories of patients who would benefit from it according to the current guidelines (in addition to women of childbearing age etc) should be better specified
This has been added on page 7, lines 208-211
4) this article should give a little more space to role of endoscopic endonasal EEA approach
I also recommend that you also refer to the fact that even more invasive tumors that move from the midline and go lateral can be attacked through combined endonasal and transorbital ETOA approaches, in this regard read and cite: https://doi.org/10.3390/jcm13092712
Thank you for the insightful comment. We have expanded on surgical approaches; please see page 7, reappraisal of surgery for prolactinomas: paragraph 2.
Reviewer 2 Report
Comments and Suggestions for Authors
This is a narrative review discussing new factors that have reconsidered the role of surgery as primary treatment in prolactinomas in certain subsets of patients, in agreement with the most recent guidelines.
The manuscript discusses all essential aspects favouring the consideration of initial surgical treatment in certain patinets with prolactinomas, outlining the advantages and very low risk of this approach in carefully selected patients.
Although the manuscript brings no novel information, in my opinion it clearly and beautifully summarizes this current trend (opposed to the frequent opinion in the past that medical treatment should almost always be the first treatment offered in prolactinomas) to reposition surgery in the complex treatment of prolactinomas.
Author Response
Reviewer 2
This is a narrative review discussing new factors that have reconsidered the role of surgery as primary treatment in prolactinomas in certain subsets of patients, in agreement with the most recent guidelines.
The manuscript discusses all essential aspects favouring the consideration of initial surgical treatment in certain patinets with prolactinomas, outlining the advantages and very low risk of this approach in carefully selected patients.
Although the manuscript brings no novel information, in my opinion it clearly and beautifully summarizes this current trend (opposed to the frequent opinion in the past that medical treatment should almost always be the first treatment offered in prolactinomas) to reposition surgery in the complex treatment of prolactinomas
Thank you for this feedback!
Reviewer 3 Report
Comments and Suggestions for Authors
The abstract provides an overview of the study but could be clearer in its presentation of the central argument. Simplify the abstract structure, especially the transition from dopamine agonist therapy (DA) to surgery. Begin with the issue (treatment of prolactinomas), then briefly outline the advances in surgery, before concluding with the role of shared decision-making. Avoid using overly complex sentences.
The manuscript occasionally uses inconsistent terminology (e.g., "microprolactinomas" vs. "macroprolactinomas" vs. "well-circumscribed macroprolactinomas"). Standardize terminology throughout the manuscript. For instance, define "well-circumscribed macroprolactinomas" early and use this consistently.
The introduction has lengthy sentences that may overwhelm the reader. Break down long sentences into shorter, more digestible ones. For example, "At presentation, they are classified as macroprolactinoma or microprolactinoma based on maximum dimensions > or < one centimeter, respectively" could be simplified.
Terms like "Knosp grade," "transsphenoidal surgery (TSS)," and "hypopituitarism" are used without explanation. While these may be familiar to an expert audience, they may confuse others. Provide definitions or brief explanations for specialized terms upon first use, such as "Knosp grade" and "transsphenoidal surgery."
The abstract ends somewhat abruptly and does not clearly outline the manuscript's central findings or conclusions. Conclude the abstract by summarizing the main takeaways—such as the evolving role of surgery in prolactinoma management and the importance of shared decision-making.
The manuscript repeats certain ideas, particularly regarding the role of DA therapy versus surgery. Condense and merge sections discussing similar points to avoid redundancy. For example, combine the discussions on the benefits and costs of long-term DA therapy into a single, cohesive argument.
Transitional elements between sections (e.g., from medical management to surgical management) are abrupt. Include clearer transition sentences that help guide the reader through the narrative. For instance, after discussing DA therapy in the introduction, you could use a sentence like, "While DA therapy has been the standard, recent advancements in surgical techniques have prompted reconsideration."
The abstract mentions improvements in surgical outcomes but lacks any specific data to substantiate this claim. Include key statistics or reference figures in the abstract that highlight the improved surgical outcomes (e.g., remission rates, complication rates).
The sentence stating that "remission rates of 91% vs. 60% for micro- and 77% vs. 43% for macroprolactinomas were reported for surgical vs. medical management, respectively" could confuse readers. Clarify if these figures represent a comparison between first-line surgery and DA therapy or between surgery and subsequent treatments after DA failure. Also, specify whether the remission rates include cases requiring additional treatment.
The manuscript mentions the need for adjuvant medical therapy post-surgery but does not expand on the reasons or the types of adjuvant therapies. Provide a more detailed discussion of the circumstances under which adjuvant therapy is required, the types of therapies (e.g., lower-dose DA), and their benefits or drawbacks.
The manuscript discusses remission and surgical outcomes but lacks emphasis on patient-centered outcomes, such as quality of life or patient satisfaction with surgery versus DA therapy. Include a section discussing the patient experience and how these factors might influence treatment decisions. For instance, how do patients perceive surgery versus long-term medication, particularly regarding side effects?
Several sections of the manuscript rely heavily on passive voice, which can make the writing seem less direct and harder to follow. Switch to active voice where possible. For example, instead of "It has been reported that DA therapy leads to tumor shrinkage," say "Studies report that DA therapy leads to tumor shrinkage."
The manuscript spends more time discussing DA therapy than surgery, even though surgery is presented as an evolving first-line option. Balance the discussion more evenly between DA therapy and surgical options. If surgery is becoming a more prominent first-line treatment, it should be discussed in greater detail than the current manuscript allows.
While the manuscript briefly mentions the complications of surgery, it does not delve deeply into potential risks or how they are managed. Provide a more thorough discussion of surgical risks, such as potential for hypopituitarism, diabetes insipidus, or long-term complications. It would also be beneficial to compare these risks with those associated with long-term DA therapy.
The manuscript emphasizes shared decision-making but does not provide a clear framework or examples of how to engage patients in these decisions. Include practical suggestions for clinicians on how to involve patients in the decision-making process, possibly offering a decision aid or describing a scenario where the choice between surgery and DA therapy is difficult.
The figure legend and references appear unfinished or incomplete (e.g., Figure 1, citations like [45]). Ensure all figure legends are complete and properly referenced. Double-check the formatting of the references to ensure that they conform to the journal's guidelines.
Author Response
Reviewer 3
The abstract provides an overview of the study but could be clearer in its presentation of the central argument. Simplify the abstract structure, especially the transition from dopamine agonist therapy (DA) to surgery. Begin with the issue (treatment of prolactinomas), then briefly outline the advances in surgery, before concluding with the role of shared decision-making. Avoid using overly complex sentences
Thank you, we have revised the abstract in accordance with these suggestions.
The manuscript occasionally uses inconsistent terminology (e.g., "microprolactinomas" vs. "macroprolactinomas" vs. "well-circumscribed macroprolactinomas"). Standardize terminology throughout the manuscript. For instance, define "well-circumscribed macroprolactinomas" early and use this consistently.
Terminology was standardized based on this comment.
The introduction has lengthy sentences that may overwhelm the reader. Break down long sentences into shorter, more digestible ones. For example, "At presentation, they are classified as macroprolactinoma or microprolactinoma based on maximum dimensions > or < one centimeter, respectively" could be simplified.
Thank you for the comment. We have revised the manuscript in accordance with this suggestion, with particular attention to the introduction.
Terms like "Knosp grade," "transsphenoidal surgery (TSS)," and "hypopituitarism" are used without explanation. While these may be familiar to an expert audience, they may confuse others. Provide definitions or brief explanations for specialized terms upon first use, such as "Knosp grade" and "transsphenoidal surgery."
The manuscript was edited to add definitions for Knosp grade, transsphenoidal surgery, and hypopituitarism.
The abstract ends somewhat abruptly and does not clearly outline the manuscript's central findings or conclusions. Conclude the abstract by summarizing the main takeaways—such as the evolving role of surgery in prolactinoma management and the importance of shared decision-making.
Thank you for the suggestion. We were unsure whether this comment pertained to the abstract or introduction- both were revised, but the suggested information was added to the conclusion of the introduction.
The manuscript repeats certain ideas, particularly regarding the role of DA therapy versus surgery. Condense and merge sections discussing similar points to avoid redundancy. For example, combine the discussions on the benefits and costs of long-term DA therapy into a single, cohesive argument.
Thank you, this paragraph in particular has been condensed.
Transitional elements between sections (e.g., from medical management to surgical management) are abrupt. Include clearer transition sentences that help guide the reader through the narrative. For instance, after discussing DA therapy in the introduction, you could use a sentence like, "While DA therapy has been the standard, recent advancements in surgical techniques have prompted reconsideration."
Thank you for the comment, the manuscript was revised with clearer transitional sentences implemented between topics.
The abstract mentions improvements in surgical outcomes but lacks any specific data to substantiate this claim. Include key statistics or reference figures in the abstract that highlight the improved surgical outcomes (e.g., remission rates, complication rates).
Data to substantiate this claim was added to the abstract.
The sentence stating that "remission rates of 91% vs. 60% for micro- and 77% vs. 43% for macroprolactinomas were reported for surgical vs. medical management, respectively" could confuse readers. Clarify if these figures represent a comparison between first-line surgery and DA therapy or between surgery and subsequent treatments after DA failure. Also, specify whether the remission rates include cases requiring additional treatment.
Thank you for the comment. This sentence was edited for clarification (page 8, lines 233,234).
The manuscript mentions the need for adjuvant medical therapy post-surgery but does not expand on the reasons or the types of adjuvant therapies. Provide a more detailed discussion of the circumstances under which adjuvant therapy is required, the types of therapies (e.g., lower-dose DA), and their benefits or drawbacks.
Thank you. This has been added on page 7-8, lines 215-220.
The manuscript discusses remission and surgical outcomes but lacks emphasis on patient-centered outcomes, such as quality of life or patient satisfaction with surgery versus DA therapy. Include a section discussing the patient experience and how these factors might influence treatment decisions. For instance, how do patients perceive surgery versus long-term medication, particularly regarding side effects?
The manuscript was revised with a paragraph on patient reported outcome measures added. Please see page 10- lines 275-287.
Several sections of the manuscript rely heavily on passive voice, which can make the writing seem less direct and harder to follow. Switch to active voice where possible. For example, instead of "It has been reported that DA therapy leads to tumor shrinkage," say "Studies report that DA therapy leads to tumor shrinkage."
Thank you for the comment. We reviewed the manuscript with attention to passive voice and revised to active voice in all amenable sentences.
The manuscript spends more time discussing DA therapy than surgery, even though surgery is presented as an evolving first-line option. Balance the discussion more evenly between DA therapy and surgical options. If surgery is becoming a more prominent first-line treatment, it should be discussed in greater detail than the current manuscript allows.
Thank you for the comment. The introduction was expanded to further discuss operative intervention and the history of surgery in pituitary surgery (pages 3&4), and the additional paragraph regarding surgical approaches was added.
While the manuscript briefly mentions the complications of surgery, it does not delve deeply into potential risks or how they are managed. Provide a more thorough discussion of surgical risks, such as potential for hypopituitarism, diabetes insipidus, or long-term complications. It would also be beneficial to compare these risks with those associated with long-term DA therapy.
Thank you for the comment. More thorough discussion was added as suggested, please see page 11, beginning at line 322.
The manuscript emphasizes shared decision-making but does not provide a clear framework or examples of how to engage patients in these decisions. Include practical suggestions for clinicians on how to involve patients in the decision-making process, possibly offering a decision aid or describing a scenario where the choice between surgery and DA therapy is difficult.
The manuscript was edited to discuss shared decision-making. Please see page 10, lines 292-301.
The figure legend and references appear unfinished or incomplete (e.g., Figure 1, citations like [45]). Ensure all figure legends are complete and properly referenced. Double-check the formatting of the references to ensure that they conform to the journal's guidelines.
Thank you for the comment. All figure legends were reviewed to ensure adherence with journal guidelines.
Round 2
Reviewer 1 Report
Comments and Suggestions for Authors
Dear authors, congratulations, all the indications have been executed correctly.
Reviewer 3 Report
Comments and Suggestions for Authors
The authors have addressed the comments.